# Second harmonic generation at a time-varying interface

Romain Tirole [1] ✉, Stefano Vezzoli[1], Dhruv Saxena [1], Shu Yang[1], T. V. Raziman[1], Emanuele Galiffi [2], Stefan A. Maier [1,3], John B. Pendry [1] & Riccardo Sapienza [1] ✉

Time-varying metamaterials rely on large and fast changes of the linear permittivity. Beyond the linear terms, however, the effect of a non-perturbative modulation of the medium on harmonic generation remains largely unexplored. In this work, we study second harmonic generation at an optically pumped time-varying interface between air and a 310 nm Indium Tin Oxide film. We observe a modulation contrast at the second harmonic wavelength up to 93% for a pump intensity of 100 GW/cm², leading to large frequency broadening and shift. We experimentally demonstrate that a significant contribution to the enhancement comes from the temporal modulation of the second order nonlinear susceptibility. Moreover, we show the frequency-modulated spectra resulting from single and double-slit time diffraction could be exploited for enhanced optical computing and sensing, enabling broadband time-varying effects on the harmonic signal and extending the application of Epsilon-Near-Zero materials to the visible range.

Time-varying metamaterials have both enabled novel wave phenomena and provided new perspectives on classic physical problems[1,2]. Time-varying photonics, where the linear susceptibility $\chi^{(1)}$ of the medium is modulated at optical frequencies, enables ultrafast switching of transmittivity or reflectivity (on the fs time scale)[3–5], frequency shifting and spectral modulation[6–8], beam-steering[9–11] and non-reciprocal devices[12]. Furthermore, concepts such as time-crystals[13], coherent wave control[14], and lasing[15] have been predicted and hold promise for experimental implementations.

Nonlinear effects such as harmonic generation, traditionally described by perturbative changes in the medium polarization, have been observed in time-varying media, for example through frequency shift of harmonic light in high-index silicon and Germanium metasurfaces[16,17]. In these systems, perturbative changes of the linear susceptibility in high-Q cavities lead to strong changes in reflectance and frequency shifts in the spectrum. This is due to the long lifetime of the probe within the metasurface, without the need for significant linear or nonlinear susceptibility changes. The use of a strong resonance engineering limits the operational bandwidth of the devices and

requires precise design and fabrication. Coherent control of SHG has also been achieved in 2D materials[18], as well as modulation of harmonic light through photocarrier excitation[19] and dark excitons[20].

As an alternative, epsilon-near-zero materials, and more particularly transparent conductive oxides[3], have emerged as promising platforms for time-modulation at near-infrared frequencies[21], by combining order-of-unity changes of their linear permittivity with an ultrafast response, close to a single optical cycle[22,23], without the need for strong photonic resonances and nanostructuring. In Indium Tin Oxide (ITO), such changes of the refractive index at these ultrafast time scales can be described by photocarrier excitation[24–26]. This has led to a surge of time-varying experiments, among which are the demonstrations of time refraction[27,28], single and double slit diffraction[5,22] and single-cycle dynamics[23].

At the harmonic level, frequency-shifted high harmonics generated in a CdO thin-film[29] have been reported and four-wave-mixing has also been shown to undergo time refraction[30,31] and diffraction[5] in pump-probe experiments with ITO and AZO. Furthermore, ITO can also be used for harmonic generation: Luk et al. showed in 2015 an

[1]Blackett Laboratory, Department of Physics, Imperial College London, London SW7 2BW, UK. [2]Photonics Initiative, Advanced Science Research Center, City University of New York, 85 St. Nicholas Terrace, 10031 New York, NY, USA. [3]School of Physics and Astronomy, Monash University, Clayton, VIC 3800, Australia. ✉e-mail: romain.tirole13@gc.cuny.edu; r.sapienza@imperial.ac.uk

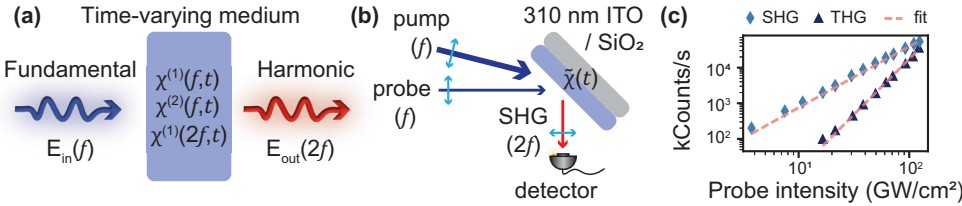

Fig. 1 | **Effects of optical modulation on SHG. a** Mechanisms for modulation of harmonic generated light. When a wave impinges on a time-varying medium, modulation in time of the linear susceptibility $\chi^{(1)}$ at the fundamental and second harmonic frequencies ($f$ and $2f$) as well as modulation of the nonlinear susceptibility $\chi^{(2)}$ can affect the generated light. **b** Diagram of the experiment: light from a probe at the fundamental frequency of 230 THz (1300 nm) generates SHG at

460 THz (650 nm) at the surface of the ITO layer, which is modulated by pumping the medium with another strong pulse at 230 THz. **c** Probe intensity dependence of the second (light blue diamonds) and third harmonic signals (dark blue triangles). Fitted power dependencies are indicated with the dashed orange line, with values at 1.72 (SHG) and 3.13 (THG).

enhancement of third harmonic generation using a plasmonic resonance[32], while Capretti et al. measured in the same year the generation of surface SHG in ITO[33], and leveraged the dependence on the normal component of the electric field for p-polarised light to enhance the SHG signal with a Berreman resonance[34]. During the review process of the present work, new results hinted at a change in the nonlinear susceptibility of ITO[35]. Yet, the effects of the non-perturbative time modulation on the nonlinear susceptibility, $\chi^{(n)}$, and the nonlinear polarization $P^{(n)}$, remain mostly unexplored.

Here, we study second harmonic generation (SHG) from a time-varying interface between an ITO thin film and air, undergoing a non-perturbative time modulation, and compare it to the modulation at the fundamental frequency. We demonstrate experimentally that the optical modulation of the linear susceptibility in ITO has a large effect on the second-order nonlinear susceptibility. Our results extend time-varying studies to the visible range and provide an interesting alternative to applications in the near-infrared, such neuromorphic optical computing, that require highly nonlinear systems, or time-diffraction, where high contrast is useful for spectroscopy.

## Results
### Concept, setup and methodology
Standard parametric nonlinear processes, e.g. second harmonic generation, are described in a perturbative framework by introducing a second order polarisation $P^{(2)}(2f)$ as a perturbative correction to the linear polarisation, which is the tensor product of the square of the fundamental field $E(f)$ at frequency $f$ and the second order nonlinear susceptibility $\chi^{(2)} = \chi^{(2)}(2f)$:

$$P^{(2)}(2f) \propto \chi^{(2)} E^2(f) \tag{1}$$

A large modulation of the linear susceptibility $\chi^{(1)}(f)$ in a time-varying medium, due to a change of the material properties typically related to photocarrier excitation in ITO[25], has a predictable effect on the field $E(f)$ which is at the basis of time-varying metamaterials[1]. Because of the quadratic dependence of the polarisation on the fundamental field $E(f)$, one should expect an enhancement of the time-varying effects on the SHG. Moreover, the temporal modulation of $\chi^{(1)}(f)$ should also induce changes in the nonlinear susceptibility $\chi^{(2)}(2f)$ and linear susceptibility at the harmonic wavelength $\chi^{(1)}(2f)$, as sketched in Fig. 1a. In the perturbative framework of nonlinear optics, these changes are normally considered negligible, compared to the modulation of $\chi^{(1)}(f)$ and its effect on the fundamental field $E(f)$.

In our experiment, we consider the SHG generated at the surface of a 310 nm layer of commercially available ITO (Präzisions Glas & Optik GmbH) by a probe beam at the fundamental frequency $f = 230$ THz, p-polarised and impinging from air at 45° incidence angle, near the Berreman resonance (see Supplementary Fig. 1 for linear properties). The interface undergoes a strong time modulation upon optical pumping by a pump pulse at 53° incidence angle ($f = 230$ THz, 225 fs

FWHM). The relative arrival time between the probe and the pump pulse is controlled via a delay stage. As illustrated in Fig. 1b, a spectrometer is then used to analyze the effect of the modulation on the reflected beam, both at the fundamental and harmonic frequencies, by using different spectral filters. The ITO thin film exhibits an ENZ frequency of 248 THz: below this frequency, the ITO layer is metallic and reflective, while above this frequency it is dielectric and transparent. As the ENZ frequency is expected to decrease under infrared pumping due to intraband transitions[25], we choose a probe frequency of 230 THz, in the metallic region. In this way, a large reflectivity drop can be observed during the metallic-to-dielectric transition under optical pumping. We can also directly measure the linear changes of the thin film at $2f$ as a reference, by using the SHG generated by a beta barium borate crystal as a probe beam.

Figure 1c shows the generated second and third harmonic signals as a function of probe intensity. The power dependence of the two signals yield fitted slopes (red dashed lines) of 1.72 and 3.13, respectively, deviating slightly from the expected values of 2 and 3 for a quadratic and cubic nonlinearity, respectively. We attribute this deviation to self-modulation of the signal in the intensity range of interest: above 20 GW/cm2 the probe induces strong enough changes in the material to modulate itself, with strong features arising above 50 GW/cm2 (see Supplementary Fig. 2). We choose to focus our study on the second harmonic generation in reflection as (1) SHG in a centrosymmetric material, such as ITO, only comes from a relatively thin region close to the surface and (2) most of the reflected beam at the fundamental frequency comes from the first air/ITO interface, due to the strong absorption of ITO at the operating frequencies leading to little second harmonic generation from the second ITO/substrate interface. This configuration simplifies the interpretation of the results, as the effect of pulse propagation and its thickness-dependent self- and cross-phase modulation can be neglected in the first approximation. Furthermore, we are focusing on SHG instead of THG as (3) high probe intensities are required to measure the THG signal, which leads to self-modulation and 4) the added processes of propagation and multiple reflections and the inhomogeneous pump modulation across the sample thickness are harder to model, THG originating from the bulk of the material rather than the surface. We also choose a low probe intensity (<5 GW/cm²) so that the generation of second harmonic in the absence of the pump is perturbative, and time-modulation comes only from the much stronger pump beam. The following results are also expected to hold for any thickness of ITO, as it relies on material modulation, though thinner samples could exhibit more complex modulations due to the SHG at the second interface and the multiple reflections of pump and SHG beams.

Our method to evaluate the nonlinear susceptibility dynamics of the material are as follow: first, by measuring the change in reflectivity of the fundamental beam from the ITO, we evaluate the modulation of linear material properties. This allows us to model through transfer matrix method (TMM) the contribution of the change in fundamental

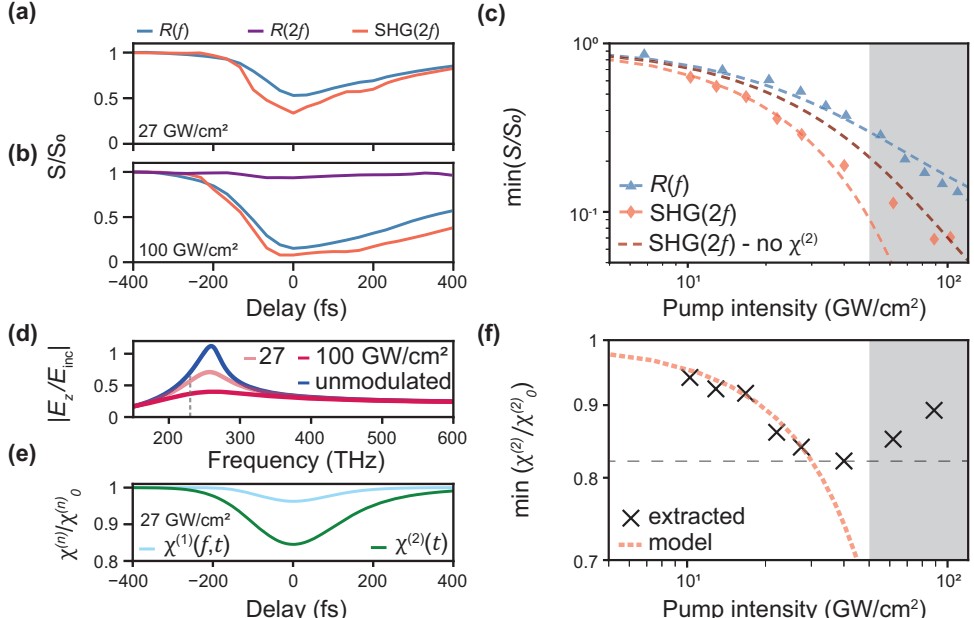

**Fig. 2 | Comparison of the modulation amplitude of fundamental and harmonic signals. a, b** Relative change $S/S_0$ in the total intensity of the reflected signal $S$ (integrated spectrum) as a function of delay, for the probe beam at the fundamental frequency $f$ (blue curve), its second harmonic at $2f$ (red curve) and another probe beam at $2f$ (purple curve), for a pump intensity of (a) 27 GW/cm² and (b) 100 GW/cm². **c** Pump intensity dependence of the maximal relative change in SHG (red diamonds) and reflected fundamental at $f$ (blue triangles), on a logarithmic scale. The agreement with the anharmonic oscillator model (dashed lines) is excellent for the fundamental, as well as for the harmonic at low intensities, when taking into account the variation in $\chi^{(2)}$ (dark red dashed line for a model without $\chi^{(2)}$ changes). The grey-shaded area indicates the saturation region. **d** Coupling

spectrum of the probe electric field to the $z$-polarised field at the surface, in the absence of pumping (blue curve) and for pump intensities of 27 GW/cm² and 100 GW/cm². The dashed grey line indicates the original carrier frequency of the probe. **e** Predicted relative change in susceptibility $\chi^{(1)}(f)$ (top, light blue) and $\chi^{(2)}$ (bottom, dark green) as a function of time for a pump intensity of 27 GW/cm² in a 310 nm film of ITO at 230 THz. **f** Maximum modulation of the nonlinear susceptibility as extracted from experiment (black crosses) against the values predicted by the anharmonic oscillator model (dashed orange line). The grey dashed line indicates the estimated maximum $\chi^{(2)}$ change of 18%. The grey-shaded area indicates the saturation region as in panel (c).

electric field to the change in SHG. By expressing the nonlinear polarization as the squared electric field normal to the interface, we can calculate the SHG field expected at the detector, bar the variations in the nonlinear susceptibility $\chi^{(2)}$. The induced modulation on $\chi^{(2)}$ can then be inferred by comparing this model with the measured SHG modulation. In order to model these experimentally measured variations, we use a simple anharmonic oscillator model and derive an evolution of the nonlinear susceptibility from the change in linear properties (see Methods). The observed changes are found to be in excellent agreement with our predictions at low to medium pump intensities, showing the efficiency of the model in capturing the physics at play despite its crudeness and multiple simplifying assumptions.

## Temporal modulation of SHG

We excite the medium with a short laser pulse and use high pump powers to approximate a step-like variation of the reflectivity, whose sign is opposite to what previously observed in the time diffraction of an ITO/Gold film[22]. The modulation of the sample reflectivity at the fundamental frequency $R(f)$ is shown in Fig. 2a, b as a function of the relative pump/probe delay (continuous blue curve), for pump intensities of (a) 27 and (b) 100 GW/cm². One can see that the modulation of the SHG (red curve) is larger than the modulation at the fundamental frequency in both cases. In contrast, the modulation experienced by a probe beam at $2f$, $R(2f)$ (purple curve, Fig. 2b), is 13 times smaller, and therefore the contribution of the temporal variation of the linear susceptibility $\chi^{(1)}(2f)$ to the SHG dynamics will be neglected in the following discussion (see Supplementary Fig. 3 for more details).

In order to compare the time-varying effect at the fundamental and harmonic frequencies, we study the reflectivity modulation depth as a function of the pump intensity, as shown in Fig. 2c. The SHG signal

(red diamonds) decreases at a higher pace with intensity than the fundamental (blue triangles), with the modulation depth saturating for intensities above 50 GW/cm², the saturation region being highlighted by the grey-shaded area (see Supplementary Fig. 4 for a linear scale). As one would expect given the quadratic dependence of the nonlinear polarisation on the fundamental fields, the SHG converges towards this saturation regime faster than its fundamental.

A simple Drude model of the ITO permittivity with a time-varying plasma frequency and electron scattering rate enables to calculate the expected change of the linear susceptibility $\chi^{(1)}(f, t)$ (see Methods). During optical pumping, intraband transitions drive the electrons in the conduction band to higher energy states, which due to the non-parabolicity of the band exhibit a different effective mass, in addition to a different scattering rate[3,24–26]. The calculated change in the Fresnel reflection coefficient of the interface as a function of pump intensity reproduces well the evolution of the modulation amplitude at the fundamental frequency (dashed blue curve). The SHG can be described as originating from a single non-linear surface polarization source, which evolves in time due to changes in the linear and nonlinear susceptibility of the material. By simply using the square of the modulated fundamental field normal to the interface $E_z(f)$ as the source of the second-order polarisation $P^{(2)}(2f)$, one falls short of reproducing the measured SHG contrast (dashed red curve in Fig. 2c).

However, when adding the contribution of a time-varying $\chi^{(2)}(t)$, the agreement with the data improves considerably (dashed orange curve in Fig. 2c). This is because the evolution of $\chi^{(1)}(f, t)$ and $\chi^{(2)}(t)$ are not independent, and modulating one always implies a modulation of the other. This is predicted using a simple anharmonic oscillator model[36], where the plasma frequency $\omega_p$ and the electron scattering

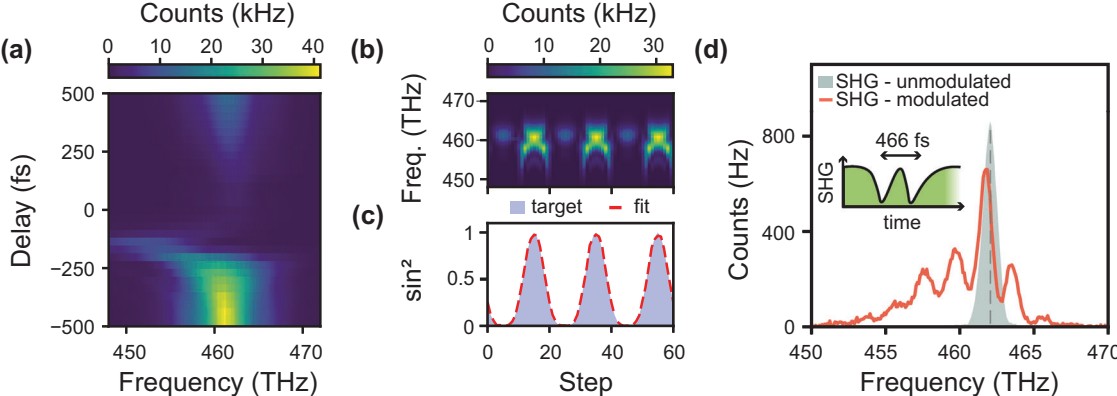

**Fig. 3 | Spectral modulation of the harmonic signal. a** SHG intensity spectra as a function of delay, for a probe duration of 225 fs and a pump intensity of 100 GW/cm². **b** Time-modulated SHG spectra against delay steps and (**c**) ridge regression (red dashed curve) to link the step input to a target function (blue-shaded area).

**d** SHG diffraction spectrum for two slits in time separated by 466 fs, for a pump intensity of 33 GW/cm². The original probe pulse, centered at 462 THz with a width of 1.12 THz, is indicated with the green-shaded area.

rate $\gamma$ are modulated (see Methods for a derivation and discussion of the model).

At first sight, one could think that a decrease in the reflected signal at the fundamental frequency would lead to an increased SHG signal, as more field would penetrate the medium and contribute to the nonlinear polarization. Yet, due to the sharp increase in electron scattering rate, the Berreman resonance is suppressed during modulation. The predicted spectrum of the probe $z$-polarised electric field to the surface of the ITO layer is shown in Fig. 2d, under modulation at pump intensities of 27 GW/cm² (pink curve) and 100 GW/cm² (red curve). In comparison to the unmodulated spectrum (blue curve), the Berreman resonance is destroyed, which leads to a reduction in surface fields at the origin of SHG (see Supplementary Fig. 5) as observed in the experiment.

Surprisingly, the model shows in Fig. 2d that the depth of the modulation of the nonlinear susceptibility $\chi^{(2)}(t)$ is expected to be larger than that of the linear susceptibility $\chi^{(1)}(f, t)$ even for a moderate intensity of 27 GW/cm². This is due to the cubic dependence of $1/\chi^{(2)}$ on the electron scattering rate, in comparison to $1/\chi^{(1)}$ which varies linearly with $\gamma$.

Furthermore, as shown in Fig. 2f, the extracted $\chi^{(2)}$ relative change with intensity (black crosses) matches very well with the anharmonic oscillator model (dashed orange line) for intensities below saturation (see Methods for details on extraction). The anharmonic model deviates from experiment at high intensities, which we attribute to higher order effects taking place in ITO as reported in literature[37]. A full modelling, requiring a higher number of fitting parameters, could potentially reveal which effects dominate at high intensities[38] Nevertheless, below pump intensities of 50 GW/cm², the data demonstrates a modulation of the $\chi^{(2)}$ that can be explained through the anharmonic oscillator model.

### Applications of time-varying second harmonic generation

Having shown an enhanced modulation depth of SHG thanks to a modulation of the second order nonlinearity of ITO at its interface, we now look into what role this new mechanism can play in the current applications brought forward by the field. We propose here to look into frequency shifting, reconstruction algorithms and ultrafast spectroscopy.

The experimental SHG spectrum carries evident signatures of the step-like temporal modulation induced by a 100 GW/cm² pump beam. As plotted in Fig. 3a, the spectrum clearly broadens and red-shifts at slightly negative delays, when the probe arrival time coincides with the pump leading edge, which corresponds to the excitation of the medium. As pointed out in previous studies of a

time-varying ITO layer[5,28], frequency shift and broadening can be respectively linked to changes in the phase and amplitude of the complex reflection coefficient of the system. Although qualitatively similar to the time-diffraction at the fundamental frequency (see Supplementary Fig. 6 for a detailed comparison), the overall shift of the harmonic spectrum is twice as large, with a much larger suppression of the peak around the unmodulated frequency. Both spectral broadening and shift increase with the magnitude of the time-modulation, driven by the pump power, although the effects saturate at about 70 GW/cm². However, the observed generation of new frequencies cannot be accurately described by our model and calls for further theoretical investigation of non-perturbative nonlinear optics, notably to include the effects of dispersion or saturation, and a more advanced material modelling. A more rigorous model such as the one presented in ref. 39 could be extended in the future to account for SHG.

Hence, the modulation of SHG signal in amplitude and frequency is very susceptible to changes in delay. This strong nonlinearity could have potential uses for optical computing and machine learning. As an example, a simple reconstruction algorithm can be used to build various waveforms such as square waves using nonlinear waveform transformation by periodically and linearly varying the delay, as illustrated in Fig. 3b (more details in Supplementary Fig. 7). The regression in Fig. 3c (red dashed curve) shows good agreement with the target shape, with a mean square error of $1.57 \times 10^{-4}$ for a $\sin^2$ shape. Thanks to the higher contrast between the modulated and unmodulated states achieved with the SHG from the suppression of the carrier frequency peak, the accessibility of efficient and affordable measurement equipment in the visible range and the overall enhanced nonlinearities, reconstructions algorithm are more interesting to implement at the harmonic rather than fundamental level. More complex transformations can also be implemented by encoding information in both space and time, such as images with pixels along axes (space, time), and measuring the SHG as a function of delay and frequency. Performing linear regression on the SHG spectrum can then allow machine learning applications such as image classification and the creation of convolutional layers.

Another potential application of the modulation of the nonlinear susceptibility in analogue computing could lie in reconfigurable image processing. For example, nonlocal metasurfaces have been used for thermal switching between edge-detection and transmission/reflection[40]. A small layer of ITO embedded in such a metasurface will couple to the field without affecting the system's resonant properties[41], enabling reconfigurable edge detection through all-optical switching of SHG.

Finally, more complex temporal modulations of SHG are possible and lead to richer spectral dynamics. As an example, we performed a double-slit diffraction experiment, similar to the one recently reported in a different ITO sample[22]. Though the physics behind the generation of the double slit interference spectrum remain the same as for the fundamental, the use of an anti-slit (decrease rather than increase of reflectivity during modulation) at the harmonic frequency has several advantages. First, whereas the peak at the carrier frequency dominates the spectrum in ref. 22, here it is suppressed, allowing for a better observation of the oscillations which is crucial in order to extract the time dynamics of the modulation. Second, as mentioned earlier, the signal can now be measured using apparatus in the visible range which are cheaper and have lower noise.

The probe pulse is now stretched to 691 fs by using a 4f system and interacts with the medium modulated by two subsequent 225 fs pump pulses, the probe arrival time is set in between the two slits' peak modulation. The reflected SHG spectrum is then measured. A clear broadening of the SHG spectrum (originally 1.12 THz FWHM, green shaded area) is observed in Fig. 3d, together with the spectral oscillations characteristic of double-slit diffraction (see Supplementary Fig. 8 for full characterisation).

The visibility of the spectral modulation, defined as (max-min)/(max+min) for the first order peak, has a constant value of $0.58 \pm 0.01$ across the range of measured slit separations, comparable to the oscillations observed for the fundamental in a previous experiment[22]. As the visibility is a measure of the coherence of the time modulation, we can conclude that all the time-varying effects leading to SHG modulation, discussed in Fig. 1a, combine in a coherent way, i.e. all new frequencies generated by the two slits at different points in time interfere coherently.

Finally, thanks to the strong suppression of the carrier frequency peak from the original spectrum in the time-modulated signal, one can also extract more information on the fast dynamic of ITO. For instance, the ratio in efficiency between the lower and higher frequency pulses which is linked to fast excitation and relaxation scales[22] is consistent with the presence of a short $\sim$ fs relaxation scale proposed by Lustig et al[23]. (see Methods and Supplementary Fig. 9). The data shown in in ref. 22 would not allow for such an estimate.

## Discussion

In conclusion, we have studied a perturbative process, SHG, in the presence of a non-perturbative modulation induced by optical intraband pumping of a time-varying ITO surface. By comparing a classic anharmonic oscillator model with experimental data, we have shown clear evidence of significant changes happening in the nonlinear susceptibility. Below saturation, these can be well explained via a simple anharmonic oscillator model. By doing so, we have decoupled the modulation on the fundamental fields generating SHG, from the induced modulation on higher order susceptibilities. Moreover, our data at high power provide a testing ground for theoretician to test more advanced models and explain the observed divergence in the non-perturbative regime. We think that all these considerations and methods are universal, and they can apply to any time-varying material platform and to any nonlinear process, such THG or FWM, although the required modelling will be more complex. As a consequence of the enhanced effective nonlinearities, time-varying effects are also amplified in the frequency domain. The strong sensitivity of the spectral features to the delay time between pump and probe can be exploited, for instance, to implement machine learning schemes or reconstruct the medium's complex time dynamics.

## Methods

### The anharmonic oscillator model

The following derivation is reproduced from Robert Boyd's Nonlinear Optics[36] where it can be found in full length. Due to the

symmetry breaking at the air/ITO interface, electrons will experience an additional asymmetric potential of the form $U(x) = 1/2ax^3$ where $x$ is the electron position. This position then follows the equation of motion:

$$\ddot{x} + 2\gamma\dot{x} + \omega_0^2 x + ax^2 = -\frac{e}{m}E(t) \tag{2}$$

where $\omega_0$ and $\gamma$ are respectively the resonant frequency and the damping term, $-e$ is the electron charge, $m$ the electron mass and $a$ a nonlinear coefficient. Assuming the nonlinear term is weak in comparison to the restoration force, one can then derive the linear susceptibility of a single Drude-Lorentz oscillator using a perturbation expansion:

$$\chi^{(1)}(\omega) = -\frac{\omega_p^2}{D(\omega)} \tag{3}$$

where $\omega_p$ is the plasma frequency of the material and $D(\omega)$ the complex denominator of the Lorentz model:

$$D(\omega) = \omega^2 - \omega_0^2 + i\gamma\omega \tag{4}$$

Then, the second order nonlinear susceptibility can be derived to be:

$$\chi^{(2)}(2\omega, \omega, \omega) = \frac{ea}{m}\frac{\omega_p^2}{D(2\omega)D(\omega)^2} \tag{5}$$

ITO follows a Drude dispersion of the form:

$$\varepsilon_r(\omega) = \varepsilon_\infty - \frac{\omega_p^2}{\omega^2 + i\gamma\omega} \tag{6}$$

which matches with the Drude-Lorentz model in the case of $\omega_0 = 0$ i.e. the absence of restoring force on the electron, which is expected when considering only carriers in the conduction band as in the Drude model. Note that the Drude-Lorentz model assumes a single oscillator, which leads to the absence of an $\varepsilon_\infty$ high-frequency permittivity constant in the model. Nevertheless, the anharmonic model informs us that the second order nonlinear susceptibility is expected to vary if the Drude parameters $\omega_p$ and $\gamma$ vary following:

$$\chi^{(2)}(2\omega, \omega, \omega) = \frac{ea}{m}\frac{\omega_p^2}{(4\omega^2 + 2i\gamma\omega)(\omega^2 + i\gamma\omega)^2} \tag{7}$$

At high intensities the model is expected to fail due to higher order processes that are not accounted for[37], but its agreement with experimental data at low intensities is remarkable. The $\chi^{(2)}$ shows a similar dependence on $\omega_p$ with $\chi^{(1)}$, but its dependence on $\gamma$ is inverse cubic in comparison, leading to increasingly larger predicted relative changes at larger intensities. Since the data on the $\chi^{(2)}$ evolution shows a saturation of the modulation above 50 GW/cm$^2$, we can assume the model is correct only at low to medium pump powers, where the modulation of $\chi^{(2)}$ is of the same order of $\chi^{(1)}$.

Note that this model is simple by design, and makes several approximations. A first one is that electrons at the surface exhibit the same effective mass and scattering rate as those in the bulk. For a change in effective mass $\delta m^*$ due to intraband transitions, it can be shown keeping $\gamma$ constant that

$$\frac{\chi^{(2)}(m^* + \delta m^*)}{\chi^{(2)}(m^*)} = \frac{1}{1 + \frac{\delta m^*}{m^*}} \tag{8}$$

thus demonstrating that changes in the susceptibility remain the same regardless of the carrier concentration used in the plasma frequency, be it bulk or surface carriers.

As second approximation is that the term $a$ in the asymmetric potential is kept constant in our model. As the interface evolves under optical pumping and the asymmetry decreases, it would be possible for $a$ to evolve in time, thus inducing further time-varying effects and further changing the value of $\chi^{(2)}$. For example, as the refractive index of ITO approaches that of air during optical pumping, the asymmetry in electronic potential is expected to reduce. Estimating $a$ and the various mechanisms behind second harmonic generation at the surface of ITO such as nonlocal and magnetic forces brought forward by Rodríguez-Suné et al[38]. will require further theoretical studies.

Also note that we approach the problem with a perturbative probe generating SHG, in a medium that evolves strongly in time. For full electromagnetic treatment, expressing the field in the medium as the sum of pump and probe, the perturbative approach would be inappropriate.

## Extraction of the nonlinear susceptibility change

Knowing the static and dynamic material changes from the changes in reflectivity of the sample, a TMM code allows the computation of the surface field $E_s$ at the air/ITO interface during modulation. Approximating the Fresnel coefficient at the harmonic to be roughly constant during modulation (see Fig. 2b), we can express the measured second harmonic field as $E_{SHG} \propto \chi^{(2)} E_s^2$. Then, for a modulation intensity $I$,

$$\frac{E_{SHG}(I)}{E_{SHG}(I=0)} = \frac{\chi^{(2)}(I)}{\chi^{(2)}(I=0)} \times \frac{E_s^{2(2)}(I)}{E_s^{2(2)}(I=0)} \tag{9}$$

$$\Longleftrightarrow \frac{\chi^{(2)}(I)}{\chi^{(2)}(I=0)} = \frac{E_s^{2(2)}(I=0)}{E_s^{2(2)}(I)} \times \sqrt{\frac{I_{SHG}(I)}{I_{SHG}(I=0)}} \tag{10}$$

where $I_{SHG}(I)/I_{SHG}(I=0)$ is the experimentally measured relative change in SHG. In Fig. 2c, f, $\min(S/S_0)$ corresponds to $I_{SHG}(I)/I_{SHG}(I=0)$ and $\min(\chi^{(2)}/\chi^{(2)}_0)$ corresponds to $\chi^{(2)}(I)/\chi^{(2)}(I=0)$.

## Fully dynamic model of the material changes

We model the change in plasma frequency $\delta_\omega$ and electron scattering rate $\delta_\gamma$ by solving the following equations:

$$\frac{d\delta_\omega}{dt} = a \times I(t) - \Gamma \delta_\omega \tag{11}$$

$$\frac{d\delta_\gamma}{dt} = b \times I(t) - \Gamma \delta_\gamma \tag{12}$$

where $a$ and $b$ are coefficients representing the efficiency of energy transfer from the pump to the electrons, $\Gamma$ a relaxation rate of the excited electrons in the conduction band (not including slower dynamics such as phonon-phonon interaction), all three fitted from experimental data, and $I(t)$ is the illuminating pulse intensity. The material properties are then expressed as

$$\omega_p(t) = \omega_p^{(0)}\left(1 + \delta_\omega(t)\right) \tag{13}$$

$$\gamma(t) = \gamma^{(0)}\left(1 + \delta_\gamma(t)\right) \tag{14}$$

where $\omega_p^{(0)}$ and $\gamma^{(0)}$ are the static plasma frequency and electron scattering coefficient and $\omega_p(t)$ and $\gamma(t)$ their modulated counterparts. Using a transfer matrix method, we can then compute the reflection of the fundamental, the surface field intensity at the origin of the second

harmonic as a function of time as well as the evolution of the nonlinear susceptibility. This all allows us to compute the relative change in SHG signal.

## Experimental setup

Degenerate pump-probe experiments are led with 225 fs pulses with tunable frequency. Beam sizes were measured using a CCD camera and compared to knife-edge measurements. Delay between pumps and probe are controlled via 2 delay stages on the probe and one of the pumps. Pumps are incident at 6° angle difference on either side of the probe, the latter being incident at 45°. Linear characterization of the sample was done using a broadband lamp and comparing the reflected signal to that of a reference silver mirror. For the double slit experiment, in order to experience the effects of both pump excitations the probe is broadened in time using a 4-f spectral filtering system. Reflected signals are sent to visible and near-infrared spectrometers. The non-degenerate measurement (pumping at frequency $f$, probing at $2f$) was done by placing a beta barium borate crystal at the focus of a telescope in the probe beam path and filtering out the fundamental frequency from the probe using a bandpass filter.

## The Fourier model for double slit diffraction

To model the change in SHG and other fundamental signals as a function of delay and intensity, the single slit aperture function is modelled as:

$$f(t) = \frac{1}{\left(1 + e^{-\alpha t}\right)\left(1 + e^{\beta t}\right)} \tag{15}$$

where $\alpha$ and $\beta$ characterize the dynamics of the excitation and relaxation of the modulation. In our experimental configuration, pump intensities of the order of 10 s of GW/cm$^2$ are sufficient to induce large spectral broadening on a probe beam at the fundamental frequency, suggesting a speeding up of the medium time response in agreement with litterature[5,23]. Because of this steepening of the response at high pump power, it is possible to talk about time-diffraction and we model the system with a fast rise time of 4.36 fs (10-90%) and a slow decay time of 615 fs. A rise time longer than 10 fs leads to fewer observable oscillations.

A Fourier transform model[22] is used to inform us on the dependence of the diffraction orders peak intensity on the dynamics of the medium. The reflection coefficient amplitude of the ITO layer is expressed as

$$r(t) = A_1 \times f\left(t - \frac{S}{2}\right) + A_2 \times f\left(t + \frac{S}{2}\right) \tag{16}$$

where $S$ is the slit separation in time, and $A_1$ and $A_2$ are negative amplitude constants fitted from experimental data and $f(t)$ has been normalised. The reflected spectrum is then expressed as $FT\left[r(t) \times E_{probe}(t)\right]$ where $E_{probe}(t)$ is the probe pulse electric field in time.

## Estimation of a fast relaxation time

The Fourier model model is used to show that the 2$^{nd}$ and higher order diffraction peaks intensity on the red/blue side respectively depend on the excitation/relaxation times as well as the amplitude in reflection coefficient of said excitation/relaxation. As can be seen in Supplementary Fig. 9, the behavior of the intensity of the 2$^{nd}$ order diffraction peak is quadratic with modulation amplitude and Gaussian (centered at 0 fs) with time scale. Writing down $a$ and $b$ as the respective amplitudes of the fast excitation and relaxation, and $\tau_r$ and $\tau_d$ their respective time scales, we can predict the relative intensities of the 2$^{nd}$ order diffraction peaks on the red and blue sides of the spectrum $I_{red}(a, \tau_r)$ and $I_{blue}(b, \tau_d)$ using the Fourier model. From our

experimental data (see Supplementary Fig. 8), we can extract $a$ and $b$ (see Supplementary Fig. 7), the excitation time $\tau_r \sim$ 1-10 fs from the spectral extent of the red oscillations[22], as well as the ratio $I_{blue}(b, \tau_d)/I_{red}(a, \tau_r)$. We can then estimate a value for $\tau_d$ from the Fourier model. As we measure $a = 0.95$ and $b = 0.36$, we compute $I_{blue}(b, \tau_d)/I_{red}(a, \tau_r) = 0.14$ for $\tau_d = \tau_r$. From our experimental data, we measured an average value of $I_{blue}/I_{red} = 0.16 \pm 0.02$, hence the simulations are in agreement with the experiment. Setting a longer relaxation time results in a decrease of $I_{blue}/I_{red}$ inconsistent with experimental measurements. Note that the Fourier model is limited as it does not include effects such as dispersion of the sample or phase shifting of the probe, and thus can only give a qualitative understanding of the dynamics of the medium.

## Data availability

The source data presented in this study have been deposited in the figshare database under accession code https://doi.org/10.6084/m9.figshare.26198594.v1.

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

## Acknowledgements

E.G. acknowledges support from the Simons Foundation (855344, E.G.). R.S., S.V., J.B.P., S.A.M. acknowledge support from UKRI (EP/V048880). JBP acknowledges support from the Gordon and Betty More Foundation. S.A.M. acknowledges the Lee-Lucas Chair in Physics.

## Author contributions

Conceptualization: R.S., S.V., R.T., E.G. Methodology: R.T., S.V., R.S. Software: R.T., S.Y., T.V.R., D.S. Investigation: R.T., S.V., S.Y., D.S., T.V.R. Visualization: R.T., R.S., S.V., S.Y., D.S., T.V.R. Funding acquisition: R.S., J.B.P., S.A.M., S.V., E.G. Project administration: R.S., J.B.P., S.A.M. Supervision: R.S., S.V. Writing—original draft: R.T., S.V., R.S. Writing—review & editing: R.T., S.V., R.S., T.V.R., J.B.P., E.G., S.A.M., S.Y.

## Competing interests

The authors declare no competing interests.
