## [Peer Review file · Nature Communications]

REVIEWER COMMENTS

Reviewer #1 (Remarks to the Author):

The authors present an exciting study of second-harmonic generation frequency modulation by a time-varying refractive index in photostimulated ITO. They provide convincing experimental evidence of frequency shifts in the generated harmonic signal near zero pump-probe delay, revealing the time-dependent nature of $\chi^{(2)}$. The modulation at the harmonic frequency is excluded through the pump-probe measurements outside of the ENZ region. The simple Fourier model explains the observed shifts and fringes well. However, I believe this paper has several deficiencies preventing publication in its current form, and it can be recommended for publication after the following issues are addressed.

1. The authors used the Drude dispersion to describe the dielectric function of their film. While this model is sufficiently precise at small intensities, the very concept of a dispersive dielectric permittivity fails when the temporal modulation is introduced because harmonic waves are no longer eigenfunctions of Maxwell's equations. Using a simplified model where only the plasma frequency and the damping factor depend on time may describe the experiment; however, the free parameters, such as photoinduced free carrier density, etc., may be erroneously determined. While this does not preclude the main observations of this mainly experimental paper, I strongly suggest revising the theoretical ansatz.

2. While the authors did observe the third harmonic generated in the ITO sample, as reported in the supplementary information, it has not been discussed in the manuscript in the context of time modulation. Does the 3rd harmonic observe similar shifts as the 2nd harmonic does? If it does not, how is this accounted for in their initial hypothesis of ENZ-driven frequency shifts?

3. In the following paper [Zubyuk et al., ACS Photonics 2022, 9, 2, 493–502], frequency shifts in optical harmonics from nanoscale optically pumped materials were observed in a similar experiment. While the present effort is sufficiently exciting and impactful in many ways, it would be helpful if the authors could clarify the novelty of their idea in the context of the publication mentioned above.

Reviewer #2 (Remarks to the Author):

In this work, a second harmonic generation has been studied at an optically pumped time-varying interface between an ITO thin film and air, undergoing a non-perturbative time modulation, and compare it to the modulation at the fundamental frequency. They demonstrate that the optical modulation of the linear susceptibility in ITO has a large effect on the second-order nonlinear susceptibility. The enhancement of the SHG modulation depth originates from a combination of the quadratic dependence of the nonlinear polarization on the fundamental fields and a strong modulation of the nonlinear susceptibility. Moreover, the spectra resulting from single and double-slit time diffraction show enhanced frequency shift, broadening and modulation depth. Below are my concerns and questions.

1. The experimental setup and the test samples are similar to those shown in previous work [17]. The innovative aspects of the experiments in this paper should be clarified more clearly. Especially in the Double-slit time diffraction experiment, all processes and phenomena are excessively similar to those in the previous work [17].

2. When the pump frequency is the same as the probe frequency, as depicted in Figure 1, the nonlinearity of ITO is modulated not only by the pump beam but also by the probe beam. How can one differentiate the contributions of the pump and probe beams to the modulation of ITO? Is it possible to use only the pump beam, without the need for the probe beam, to realize the second-harmonic generation (SHG) phenomena? Does the magnitude differences of the pump and probe beams affect the conclusions of this study?

3. The abbreviation ENZ must be spelled out in full upon its first occurrence.

4. The similar phenomena presented in Fig.3a have already been studied in the previous work [8]. Please clarify the novelty of this paper.

5. It was claimed that "An enhancement of the SHG modulation depth originates from a combination of the quadratic dependence of the nonlinear polarization on the fundamental fields and a strong modulation of the nonlinear susceptibility." Please detail how, in practical measurements and theoretical analysis, one can distinguish how many contributions attributed from "nonlinear polarization on the fundamental fields", and how many contributions attributed from "strong modulation of the nonlinear susceptibility."

6. It was stated that "However, the observed generation of new frequencies cannot be accounted by our model and calls for further theoretical investigation of non-perturbative nonlinear optics".

Please detail the specific reasons behind this limitation, and clarify the applicability and universality of the conclusions given in this paper.

7. "Second-harmonic generation (SHG) in a centrosymmetric material, such as ITO, originates only from a relatively thin region near the surface." Therefore, SHG in ITO has two sources: one part comes from the intrinsic nonlinear surface susceptibility, and the other from the time-varying modulation. How can one differentiate between these two contributions in the experimental measurements and theoretical analysis?

8. To ensure that "most of the reflected beam at the fundamental frequency comes from the first air/ITO interface, ... , most of the reflected beam at the fundamental frequency comes from the first 124 air/ITO interface", the ITO layer must be sufficiently thick. Please detail the experimental fabrication process of the sample, how to ensure that the thickness of the sample processing is indeed 310nm, as well as the size of the sample area and the yield of the fabricated samples. Please also discuss the impact of different ITO thicknesses on the conclusions of this paper.

9. The concept of "SHG with time-varying metamaterials" has already been presented much earlier (<https://arxiv.org/abs/2001.03036>); and many researches on SHG and higher-harmonic generations with high efficiencies by using time-coding and space-time-coding metasurfaces and their applications in wireless communications have been presented, both in theoretical studies and experiments. The authors should compare and contrast to clarify the contribution and novelty of this paper.

Reviewer #3 (Remarks to the Author):

In the article "Second Harmonic Generation at a Time Varying Interface" authors discuss the effect of nonperturbative nonlinearities on the second order electric susceptibility $\chi^{(2)}$ of an indium doped tin oxide film. A degenerate pump-probe experiment is conducted with varying delay between pump and probe beams and observe the strength of the second harmonic intensity. It is concluded that the change in the intensity of the second harmonic beam cannot be fully captured by the change in the linear field, therefore the strong modulation must also be caused by a change in the nonlinear susceptibility parameters. Although potentially an interesting study to understand the dynamic properties of the ITO film, the paper needs better explanations and also needs to emphasize the significance of the results more clearly. There are also several critical issues that must be checked before publication. At the current stage, the manuscript is not ready for publication. Below are some major and minor comments.

1) In the introduction, the authors state, “the spectra resulting from single and double-slit time diffraction show significantly enhanced ... modulation depth, when compared to the infrared fundamental beam...”. However, on page 7 line 232-235 they say it is comparable. Which is it, significantly enhanced or comparable?

2) Authors infer from the experiment that the pumping causes a change in the nonlinear susceptibility of the material. They also show that this change is in agreement with the anharmonic oscillator model. However, it is necessary to explain or comment on what is the physical mechanism behind this change. For example, on page 6, it is suggested the material properties play a role, however, the plasma frequency enters $\chi^{(1)}$ and $\chi^{(2)}$ in the same way and potentially has the same effect. Where does the change in $\chi^{(2)}$ (double of $\chi^{(1)}$) come from? How does changing the parameters (a , b , τ_r and τ_d) affect the nonlinearity? Also, what is the physical interpretation of this change for ITO in terms of the electron movement in the bands?

3) The discussion on the Berreman mode should be made more clear in the main text. In my understanding the unmodulated medium is probed away from the ENZ regime and when it is modulated the ENZ point redshifts and coincides with the probe wavelength, therefore exciting the Berreman resonance (This should be highlighted better in the paper maybe with a plot showing modulated and unmodulated electric permittivity). Hence the reflected intensity significantly drops. It is counter intuitive to me why would this cause a decrease in the generated nonlinear signal. Although there is a brief explanation in the paper about the decreased intensity of light on the surface, there are many studies in literature showing enhanced nonlinearities at ENZ regime¹⁻³ (one specifically for enhancement of SHG from Berreman resonance)¹ so exciting Berreman resonance should be a cause of enhanced SHG. The initial probe is not at the Berreman mode and pumping moves it towards the Berreman mode. Wouldn't that increase the SHG?

4) The plots in figure 2 a indeed shows a discrepancy between reflected linear intensity and the SHG. But the difference between the two plots appears to be insufficient to account for the significant change in $\chi^{(2)}$ in figure 2c. This is probably a consequence of how the data is visualized, but it would be beneficial to see this relation between plots highlighted better and the fitting of $\chi^{(2)}$ to obtain such a trend.

5) On page 6, line 170 it is stated that difference in modulation between fundamental and second harmonic is larger at low powers. Figure 2b seems to show a larger difference at higher powers.

6) On page 4, the first order $\chi^{(1)}$ is a function of both frequency and time. I understand this to mean that formally, we average over many periods to describe the frequency dependence in time (slowly varying approximation) and therefore our time resolution is much greater than a period (~ 4 fs). In the supplement, the rise time of the model is on the order of 1-10 fs. These two points seem to contradict each other.

7) There is very little discussion about the technological implications of the study. There is a proposed reconstruction algorithm, but it is mentioned in passing with sparse explanation. Additionally, is the $\chi^{(2)}$ modulation necessary in this application. Wouldn't the modulation of the linear field stemming from the strong modulation create the same potential for such applications?

8)The inset of Figure 3d looks like there are three slits/pulses. Perhaps the left-most and right-most green boxes should be extended.

9)It would be very interesting to have a FROG trace or autocorrelation of the emitted SHG signal showing the temporal structure.

10)On page 8, it says the unmodulated SHG is suppressed. Based on figure 2b it seems modulation suppresses the SHG.

11)In the supplementary there is the high pump powers saturate the system within several femtoseconds resulting in an ultrafast rise time based on reference 5. First, we note that the intensities in reference 5 were 10 times that of this study. Is the 4.36 fs rise time still appropriate? Also, after that rise time the system continues to be pumped. One would then expect a plateau in the material response during the rest of the pulse (>200 fs). This is not captured by the model. Could excited state absorption also play a role here?

12)The last sentence of page 2 states the frequency shift and broadening of SHG is not explained by the model. I am not able to find in the figures or text where this is discussed. Perhaps I missed it.

13)The SHG from the probe beam is perturbative due to the low intensity of the probe in the unmodulated medium. However, the field in the modulated medium is much larger due to the enhancement of field with Berreman mode. In this case can the SHG still be expanded with perturbation theory. Is $\chi^{(2)} \cdot E \ll \chi^{(1)}$?

Some minor comments

14)page 2 line 46, should there be a hyphen on Tin-Oxide? Should there also be one in the abstract?

15)Page 2 line 53, silicon should not be capitalized.

16)The word “non-perturbative” on line 131 seems to be a typo. I understood this sentence to mean the probe does not perturb the system. In nonlinear optics, non-perturbative has a different connotation.

17)On page 7, lines 215-216, it is unclear to me if the terms amplitude and frequency describe the pump or the probe of the SHG signal.

18)Page 7 line 227 is missing a verb between time and set.

19)The figure S6 is labeled as S5.

20)The colors on the x axis tick marks for Figure S5 should be switched.

21)Are the alpha and beta in the definition of $f(t)$ the same as τ_r and τ_d ?

22)In the fitting of the fast relaxation time, is the result unique?

1. Passler, N. C. et al. Second Harmonic Generation from Phononic Epsilon-Near-Zero Berreman Modes in Ultrathin Polar Crystal Films. ACS Photonics 6, 1365–1371 (2019).

2. Alam, M. Z., De Leon, I. & Boyd, R. W. Large optical nonlinearity of indium tin oxide in its epsilon-near-zero region. *Science* (1979) 352, 795–797 (2016).

3. Capretti, A., Wang, Y., Engheta, N. & Dal Negro, L. Comparative Study of Second-Harmonic Generation from Epsilon-Near-Zero Indium Tin Oxide and Titanium Nitride Nanolayers Excited in the Near-Infrared Spectral Range. *ACS Photonics* 2, 1584–1591 (2015).

Replies to the referees will be marked in purple, and changes in the manuscript in red.

REVIEWER COMMENTS

Reviewer #1 (Remarks to the Author):

We thank the reviewer for the constructive comments, we have addressed here below point by point.

1. The authors used the Drude dispersion to describe the dielectric function of their film. While this model is sufficiently precise at small intensities, the very concept of a dispersive dielectric permittivity fails when the temporal modulation is introduced because harmonic waves are no longer eigenfunctions of Maxwell's equations. Using a simplified model where only the plasma frequency and the damping factor depend on time may describe the experiment; however, the free parameters, such as photoinduced free carrier density, etc., may be erroneously determined. While this does not preclude the main observations of this mainly experimental paper, I strongly suggest revising the theoretical ansatz.

A detailed ab-initio model of time-modulation in ITO is indeed still a challenge, due to the complexity of modelling the material and the optical response. We do not aim at developing new theory here, but rather aim at reproducing the experimental observations with the simplest accepted model.

Our model is indeed approximated, for example neglecting dispersion, but given the relatively small bandwidth of our pulses, we believe that dispersion does not play an important role, as in previous experiments [10.1103/PhysRevApplied.18.054067]. The material response is also simplified, using only a plasma frequency and the damping factor depend on time, which is the most common and accepted model in the community so far [10.1103/PhysRevApplied.11.064062].-The resulting time-varying linear susceptibility $\chi^{(1)}$ describes very well the evolution of the reflectivity modulation of the IR probe with pump power [Fig. 2(c)], and therefore it is what we use here. We have discussed the limitations of our theoretical model:

However, the observed generation of new frequencies cannot be accurately described by our model and calls for further theoretical investigation of non-perturbative nonlinear optics, notably to include the effects of dispersion or saturation, and a more advanced material modelling. A more rigorous model such as the one presented in ³⁵ could be extended in the future to account for SHG.

2. While the authors did observe the third harmonic generated in the ITO sample, as reported in the supplementary information, it has not been discussed in the manuscript in the context of time modulation. Does the 3rd harmonic observe similar shifts as the 2nd harmonic does? If it does not, how is this accounted for in their initial hypothesis of ENZ-

driven frequency shifts?

The THG also undergoes rich dynamics, with strong frequency shifts, as shown in the figure below.

However, for the low probe powers that we use here, low enough to avoid probe self-modulation, the THG signal is orders of magnitude smaller than the SHG (as shown in the new panel, now added to Fig. 1.c, which makes the experiments more challenging. Moreover, since THG is a volume effect, the need to include propagation, multiple reflections and the effects of inhomogeneous pump modulation across the sample thickness, makes the temporal dynamics of the THG much more complex to model, and as a result it becomes more challenging to infer the time evolution of $\chi^{(3)}$. Therefore, we have left this for further experiments. The following sentence has been added to the text on page 4:

Furthermore, we are focusing on SHG instead of THG as 3) high probe intensities are required to measure the THG signal, which leads to self-modulation and 4) the added processes of propagation and multiple reflections and the inhomogeneous pump modulation across the sample thickness are harder to model, THG originating from the bulk of the material rather than the surface.

The power dependence of SHG and THG have been moved from supplementary to Fig. 1, in order to better motivate our choice to focus on $\chi^{(2)}$ on page 3:

Figure 1. [...] (c) Probe intensity dependence of the second (light blue diamonds) and third harmonic signals (dark blue diamonds). Fitted power dependences are indicated with the dashed orange line, with values at 1.72 (SHG) and 3.13 (THG).

and is discussed in the text on page 4:

Fig. 1(c) shows the generated second and third harmonic signals as a function of probe intensity. The power dependence of the two signals yield fitted slopes (red dashed lines) of 1.72 and 3.13 respectively, deviating slightly from the expected values of 2 and 3 for a quadratic and cubic nonlinearity, respectively. We attribute this deviation to self-modulation of the signal in the intensity range of interest: above 20 GW/cm² the probe induces strong enough changes in the material to modulate itself, with strong features arising above 50 GW/cm² (see Supplementary Figure 2).

3. In the following paper [Zubyuk et al., ACS Photonics 2022, 9, 2, 493–502], frequency shifts in optical harmonics from nanoscale optically pumped materials were observed in a similar experiment. While the present effort is sufficiently exciting and impactful in many ways, it would be helpful if the authors could clarify the novelty of their idea in the context of the publication mentioned above.

The switching of harmonic generation in high-index cavities (high-Q metasurfaces in ACS Photonics 2022, 9, 2, 493) leverages physical mechanisms that are different than time-modulation in ITO. In cavities, the refractive index is modulated very weakly ($\sim 2 \times 10^{-3}$ for 100 GW/cm² [10.1126/sciadv.aaw3262]), as a small perturbation and switching is enabled by the long lifetime and long probe interaction in the cavity. These systems exhibit no significant change in nonlinear susceptibility. Instead, in the ITO interface with a low Q-factor (~ 6 from TMM, see Supp. Fig. 1(c)) the material time-modulation induces strong changes of linear (\sim unity) and nonlinear susceptibility. The following paragraph has been rewritten in the introduction to highlight the novelty of our work, page 2:

In these systems, perturbative changes of the linear susceptibility in high-Q cavities lead to strong changes in reflectance and frequency shifts in the spectrum. This is due to the long lifetime of the probe within the metasurface, without the need for significant linear or nonlinear susceptibility changes. The use of a strong resonance engineering limits the operational bandwidth of the devices and requires precise design and fabrication.

Reviewer #2 (Remarks to the Author):

We thank the Reviewer for the detailed analysis of our paper. We have addressed all points raised here below.

1. The experimental setup and the test samples are similar to those shown in previous work [17]. The innovative aspects of the experiments in this paper should be clarified more clearly. Especially in the Double-slit time diffraction experiment, all processes and phenomena are excessively similar to those in the previous work [17].

Ref 17 describes our previous study of the linear modulation of material, aimed at the study of the temporal analogue of the double-slit diffraction. This manuscript is about the study of the nonlinear modulation, while the double-slit diffraction of the SHG signal, never reported before, is just an example and not the main result of the manuscript. The key innovation of

our work is the modulation of the nonlinear system response without relying on nano-photonics engineering of long-lived states. Moreover, we show that the nonlinear susceptibility $\chi^{(2)}$ is time-modulated, which is an important step beyond linear time-modulation $\chi^{(1)}$.

Double-slit diffraction of SHG comes also with advantages over the linear counterpart, that we have described on page 9:

Though the physics behind the generation of the double slit interference spectrum remain the same as for the fundamental, the use of an anti-slit (decrease rather than increase of reflectivity during modulation) at the harmonic frequency has several advantages. First, whereas the peak at the carrier frequency dominates the spectrum in 19, here it is suppressed, allowing for a better observation of the oscillations which is crucial in order to extract the time dynamics of the modulation. Second, as mentioned earlier, the signal can now be measured using apparatus in the visible range which are cheaper and have lower noise.

2. When the pump frequency is the same as the probe frequency, as depicted in Figure 1, the nonlinearity of ITO is modulated not only by the pump beam but also by the probe beam. How can one differentiate the contributions of the pump and probe beams to the modulation of ITO? Is it possible to use only the pump beam, without the need for the probe beam, to realize the second-harmonic generation (SHG) phenomena? Does the magnitude differences of the pump and probe beams affect the conclusions of this study?

If brought to higher intensities than what we use here, the probe can also modulate its own spectrum and amplitude through self-modulation. In our experiments, we keep the probe intensity very low, around 1 GW/cm², in order to avoid any of those contributions. This is mentioned on page 4 and the self-modulated spectrum of the probe in the absence of the pump is now shown in supplementary figure 2(c) page 19 (see below). A power dependence shows clearly that self-modulation occurs only at high-enough powers, larger than 20-50 GW/cm².

Supplementary Figure 2. [...] (c) Self-modulation of the probe: carrier frequency shift (cyan diamonds) and pulse bandwidth broadening (orange diamonds) as a function of probe intensity, in the absence of the pump.

Regarding the second part of the question, a single pump beam generates second harmonic, which can self-modulate at high-intensities just as for the probe. For example,

self-modulation of harmonic has been observed in a CdO thin film for high pump-intensities [10.1038/s41567-019-0584-7]. The magnitude difference between pump and probe is always crucial in pump-probe experiments, as the measurement relies on the probe not inducing modulations. However, the main conclusion, which is the demonstration of a temporal modulation of the $\chi^{(2)}$, is general and holds regardless of the ratio of intensities being used.

The power dependence of unmodulated SHG has now been moved to Fig. 1 and is discussed in the text as follows on pages 3 and 4:

Figure 1. [...] (c) Probe intensity dependence of the second (light blue diamonds) and third harmonic signals (dark blue diamonds). Fitted power dependences are indicated with the dashed orange line, with values at 1.72 (SHG) and 3.13 (THG).

Fig. 1(c) shows the generated second and third harmonic signals as a function of probe intensity. The power dependence of the two signals yield fitted slopes (red dashed lines) of 1.72 and 3.13 respectively, deviating slightly from the expected values of 2 and 3 for a quadratic and cubic nonlinearity, respectively. We attribute this deviation to self-modulation of the signal in the intensity range of interest: above 20 GW/cm² the probe induces strong enough changes in the material to modulate itself, with strong features arising above 50 GW/cm² (see Supplementary Figure 2).

The power difference of pump and probe enables us to have strong modulations (driven by the pump) while the low probe intensity ensures that SHG is within the perturbative regime, and without self-modulation to complicate the analysis.

3. The abbreviation ENZ must be spelled out in full upon its first occurrence.

Fixed.

4. The similar phenomena presented in Fig.3a have already been studied in the previous work [8]. Please clarify the novelty of this paper.

This work's main novelty is the modulation of the nonlinear system response without relying on nano-photonic engineering of long-lived states. Moreover, we show that also the nonlinear susceptibility is time-modulated, which is an important step beyond linear time-modulation.

The switching of harmonic generation in high-index cavities (high-Q metasurfaces in ACS Photonics 2022, 9, 2, 493) leverages physical mechanisms that are different than time-

modulation in ITO. In cavities, the refractive index is modulated very weakly ($\sim 2 \times 10^{-3}$ for 100 GW/cm² [10.1126/sciadv.aaw3262]), as a small perturbation and switching is enabled by the long lifetime and long probe interaction in the cavity. These systems exhibit no significant change in nonlinear susceptibility. Instead, in the ITO interface with a low Q-factor (~ 6 from TMM, see Supp. Fig. 1(c)) the material time-modulation induces strong changes of linear (\sim unity) and nonlinear susceptibility. The following paragraph has been rewritten in the introduction to highlight the novelty of our work, page 2:

In these systems, perturbative changes of the linear susceptibility in high-Q cavities lead to strong changes in reflectance and frequency shifts in the spectrum. This is due to the long lifetime of the probe within the metasurface, without the need for significant linear or nonlinear susceptibility changes. The use of a strong resonance engineering limits the operational bandwidth of the devices and requires precise design and fabrication.

The abstract, introduction and conclusion as a whole have been rewritten for the same clarity purpose.

5. It was claimed that "An enhancement of the SHG modulation depth originates from a combination of the quadratic dependence of the nonlinear polarization on the fundamental fields and a strong modulation of the nonlinear susceptibility." Please detail how, in practical measurements and theoretical analysis, one can distinguish how many contributions attributed from "nonlinear polarization on the fundamental fields", and how many contributions attributed from "strong modulation of the nonlinear susceptibility."

We have modified the manuscript to better reflect and break down the logical steps in our measurement and analysis that allow us to estimate these contributions, page 5:

Our method to evaluate the nonlinear susceptibility dynamics of the material are as follow: first, by measuring the change in reflectivity of the fundamental beam from the ITO, we evaluate the modulation of linear material properties. This allows us to model through transfer matrix method (TMM) the contribution of the change in fundamental electric field to the change in SHG. By expressing the nonlinear polarization as the squared electric field normal to the interface, we can calculate the SHG field expected at the detector, bar the variations in the nonlinear susceptibility $\chi^{(2)}$. The induced modulation on $\chi^{(2)}$ can then be inferred by comparing this model with the measured SHG modulation. In order to model these experimentally measured variations, we use a simple anharmonic oscillator model and derive an evolution of the nonlinear susceptibility from the change in linear properties (see Methods). The observed changes are found to be in excellent agreement with our predictions at low to medium pump intensities, showing the efficiency of the model in capturing the physics at play despite its crudeness and multiple simplifying assumptions.

6. It was stated that "However, the observed generation of new frequencies cannot be accounted by our model and calls for further theoretical investigation of non-perturbative nonlinear optics". Please detail the specific reasons behind this limitation, and clarify the applicability and universality of the conclusions given in this paper.

Spectral modulation of SHG is difficult to model due to the requirement of having a precise knowledge of the evolution of the phase and amplitude evolution inside and outside the

material on a time scale inaccessible from the pump-probe measurement [10.1103/PhysRevLett.130.203803, 10.1103/PhysRevApplied.18.054067]. The modelling of the time-dynamics is not fully understood at the fundamental frequency and will be further difficult to model at the harmonic, which would require additional fitting parameters which are difficult to determine [10.1364/OL.42.003225, 10.1063/1.5129627].

For these reasons, we have chosen a simple Lorentz oscillator model in order to accurately capture the physics in the modulation of the nonlinear susceptibility, without aiming at reproducing spectral changes. This does not affect the main result of the paper, which is the modulation of nonlinear susceptibility and the useful added nonlinearity of SHG. These results are expected to apply to other materials as well as other nonlinear processes (e.g. THG). This is because they describe a fundamental material property, the change of the nonlinear susceptibility under optical pumping, whereas in comparison frequency shifting also depends on dispersion engineering, material geometry and many more factors.

The following sentence has been added to the text on page 8:

However, the observed generation of new frequencies cannot be accurately described by our model and calls for further theoretical investigation of non-perturbative nonlinear optics, notably to include the effects of dispersion or saturation, and a more advanced material modelling. A more rigorous model such as the one presented in ³⁵ could be extended in the future to account for SHG.

and in the conclusion on page 10:

we have shown clear evidence of significant changes happening in the nonlinear susceptibility. Below saturation, these can be well explained via a simple anharmonic oscillator model. By doing so, we have decoupled the modulation on the fundamental fields generating SHG, from the induced modulation on higher order susceptibilities. Moreover, our data at high power provide a testing ground for theoretician to test more advanced models and explain the observed divergence in the non-perturbative regime. We think that all these considerations and methods are universal, and they can apply to any time-varying material platform and to any nonlinear process, such THG or FWM, although the required modelling will be more complex.

7. "Second-harmonic generation (SHG) in a centrosymmetric material, such as ITO, originates only from a relatively thin region near the surface." Therefore, SHG in ITO has two sources: one part comes from the intrinsic nonlinear surface susceptibility, and the other from the time-varying modulation. How can one differentiate between these two contributions in the experimental measurements and theoretical analysis?

SHG should be perceived as coming from a single source at the interface, which evolves in time due to changes in the permittivity. The time-modulation is not a source of SHG, it induces modulations of SHG generated by the layer. The following text has been added on page 6:

The SHG can be described as originating from a single nonlinear surface polarization source, which evolves in time due to changes in the linear and nonlinear susceptibility of the material.

8. To ensure that "most of the reflected beam at the fundamental frequency comes from the first air/ITO interface, ... , most of the reflected beam at the fundamental frequency comes from the first 124 air/ITO interface", the ITO layer must be sufficiently thick. Please detail the experimental fabrication process of the sample, how to ensure that the thickness of the sample processing is indeed 310nm, as well as the size of the sample area and the yield of the fabricated samples. Please also discuss the impact of different ITO thicknesses on the conclusions of this paper.

The sample was purchased from PGO, and matches ellipsometric measurements. The following comment has been added on page 4:

In our experiment, we consider the SHG generated at the surface of a 310 nm layer of commercially available ITO (Präzisions Glas & Optik GmbH).

A discussion on the role of thickness of ITO has been added to the text on page 4:

The following results are also expected to hold for any thickness of ITO, as it relies on material modulation, though thinner samples could exhibit more complex modulations due to the SHG at the second interface and the multiple reflections of pump and SHG beams.

9. The concept of "SHG with time-varying metamaterials" has already been presented much earlier (<https://arxiv.org/abs/2001.03036>); and many researches on SHG and higher-harmonic generations with high efficiencies by using time-coding and space-time-coding metasurfaces and their applications in wireless communications have been presented, both in theoretical studies and experiments. The authors should compare and contrast to clarify the contribution and novelty of this paper.

The reference [Arxiv:2001.03036](https://arxiv.org/abs/2001.03036) uses a technique known as EFISH, where a DC field breaks the crystal symmetry and enables SHG from a third order nonlinear process. This is built on very different physics as it does not rely on a fast modulation of the optical constant. As a consequence, dynamics are much slower, the switching speed of the DC field being far below the femtosecond scaled required for time-varying physics. As such we believe it is a very different process.

The space time coding metasurfaces commonly found in literature also rely on electronic switching speeds, instead of nonlinear optics, on timescales from the ms to ns, much slower than the fs scale presented here. The only result of fast switching reported so far in optics [10.1126/sciadv.adf1015] leverages a thick slab of silica (100 microns vs 310 nm for our film) driven by very short pulses (2.7 fs vs our 225 fs) to achieve much smaller reflectivity changes (8.4×10^{-3} at 265 GW/cm² vs 4.2×10^{-1} at 100 GW/cm²). While coding metasurfaces is an exciting field, relying on reconfiguration, it is very different in nature to is what reported here. The ultrafast modulation of the nonlinear susceptibility in Indium Tin Oxide demonstrates its potential for space-time encoding metasurfaces.

Reviewer #3 (Remarks to the Author):

We thank the reviewer for the useful comments and needs for emphasizing the significance of our work. We detail our reply here below.

1) In the introduction, the authors state, “the spectra resulting from single and double-slit time diffraction show significantly enhanced ... modulation depth, when compared to the infrared fundamental beam...”. However, on page 7 line 232-235 they say it is comparable. Which is it, significantly enhanced or comparable?

On page 7, it is the visibility of the diffracted peaks that are indeed comparable when compared to the infrared, which means the SHG diffraction process is as coherent as the fundamental. The visibility being defined as a ratio in intensity between a peak (here 1st order on the red end of the spectrum) and the next dip, it is not directly related to modulation depth. The modulation depth presented in the introduction is defined as the maximum achievable modulation on the integrated spectrum, and does not relate to the visibility of the oscillations. The introduction and abstract have been rewritten and this ambiguity has been removed.

2) Authors infer from the experiment that the pumping causes a change in the nonlinear susceptibility of the material. They also show that this change is in agreement with the anharmonic oscillator model. However, it is necessary to explain or comment on what is the physical mechanism behind this change. For example, on page 6, it is suggested the material properties play a role, however, the plasma frequency enters $\chi^{(1)}$ and $\chi^{(2)}$ in the same way and potentially has the same effect. Where does the change in $\chi^{(2)}$ (double of $\chi^{(1)}$) come from? How does changing the parameters (a , b , τ_r and τ_d) affect the nonlinearity? Also, what is the physical interpretation of this change for ITO in terms of the electron movement in the bands?

We thank the referee for the suggestion to clarify our theoretical model. The referee is right to point out that $\chi^{(1)}$ and $\chi^{(2)}$ are expected to follow the same trend at low power and thus their expected modulation should be comparable. However, we should keep in mind that γ also changes strongly, and that $\chi^{(2)} \propto (1/\gamma^3)$ shows a stronger dependence than the $\chi^{(1)} \propto (1/\gamma)$. The following sentence has been added on page 16:

The $\chi^{(2)}$ shows a similar dependence on ω_p with $\chi^{(1)}$, but its dependence on γ is inverse cubic in comparison, leading to increasingly larger predicted relative changes at larger intensities. Since the data on the $\chi^{(2)}$ evolution shows a saturation of the modulation above 50 GW/cm², we can assume the model is correct only at low to medium pump powers, where the modulation of $\chi^{(2)}$ is of the same order of $\chi^{(1)}$.

The time constants do not affect the strength of the modulation and only intervene when considering the double slit experiment. Coefficients a and b impact the amplitude of the reflectivity modulation $r(t)$: a negative corresponds to intraband pumping changing the

effective mass due to the non-parabolicity of the band, lowering reflectivity, while b positive means an increase in loss within the material due to the higher energy of the electrons in the conduction band. The changes of plasma frequency and scattering rate have been well studied in references [20,21].

The abstract has been changed to reflect better the results:

The measured change in nonlinear susceptibility matches well with an anharmonic oscillator model at low to medium intensities, while at higher intensities higher-order effects saturate the modulation. [...] Enabling and strengthening broadband time-varying effects on the harmonic signal extends the application of materials to the visible range and calls for further theoretical exploration of non-perturbative nonlinear optics at high intensities.

Also, what is the physical interpretation of this change for ITO in terms of the electron movement in the bands?

The following sentence has been added to the text on page 6 to further explain the movement of electrons in the band:

During optical pumping, intraband transitions drive the electrons in the conduction band to higher energy states, which due to the non-parabolicity of the band exhibit a different effective mass and scattering rate ^{3,21–23}.

3) The discussion on the Berreman mode should be made more clear in the main text. In my understanding the unmodulated medium is probed away from the ENZ regime and when it is modulated the ENZ point redshifts and coincides with the probe wavelength, therefore exciting the Berreman resonance (This should be highlighted better in the paper maybe with a plot showing modulated and unmodulated electric permittivity). Hence the reflected intensity significantly drops. It is counter intuitive to me why would this cause a decrease in the generated nonlinear signal. Although there is a brief explanation in the paper about the decreased intensity of light on the surface, there are many studies in literature showing enhanced nonlinearities at ENZ regime^{1–3} (one specifically for enhancement of SHG from Berreman resonance)¹ so exciting Berreman resonance should be a cause of enhanced SHG. The initial probe is not at the Berreman mode and pumping moves it towards the Berreman mode. Wouldn't that increase the SHG?

The decrease in electric field is indeed counter-intuitive, and it is explained by the fact that the Berreman mode is suppressed during modulation due to the increase in losses in our system. In unmodulated systems, the losses remain low and indeed SHG and THG are maximal at the Berreman resonance due to field enhancement. This can be seen in Supp. Fig. 4(a): upon modulation, the Berreman resonance mostly disappears, hence no enhancement of the electric field is observed and the SHG drops. The panel has been adapted and added to Fig. 2 on page 5

Figure 2. [...] (d) Coupling spectrum of the probe electric field to the z-polarised field at the surface, in the absence of pumping (blue curve) and for pump intensities of 27 GW/cm² and 100 GW/cm². The dashed grey line indicates the original carrier frequency of the probe. [...]

and discussed on page 7:

Yet, due to the sharp increase in electron scattering rate, the Berreman resonance is suppressed during modulation. The predicted spectrum of the probe z-polarised electric field to the surface of the ITO layer is shown in Fig. 2(d), under modulation at pump intensities of 27 GW/cm² (pink curve) and 100 GW/cm² (red curve). In comparison to the unmodulated spectrum (blue curve), the Berreman resonance is destroyed, which leads to a reduction in surface fields at the origin of SHG (see Supplementary Figure 5) as observed in the experiment.

4) The plots in figure 2 a indeed shows a discrepancy between reflected linear intensity and the SHG. But the difference between the two plots appears to be insufficient to account for the significant change in $\chi^{(2)}$ in figure 2c. This is probably a consequence of how the data is visualized, but it would be beneficial to see this relation between plots highlighted better and the fitting of $\chi^{(2)}$ to obtain such a trend.

We thank the referee for this very important suggestion. As a result, we have improved our analysis of the data and produced a new version of Fig.2, which we hope is now clearer to interpret. The nonlinear susceptibility has now been extracted from our data using the equations shown in the Methods section (see below) as a function of pump intensity, and is shown next to our model in panel (f) in the new version of Fig. 2 on page 5. A change to a logarithmic scale highlights the difference better in panel (c), indeed as all signals get closer to zero it is harder to distinguish them on a linear scale.

Figure 2. [...] (c) Pump intensity dependence of the maximal relative change in SHG (red diamonds) and reflected fundamental at f (blue diamonds), on a logarithmic scale. The agreement with the anharmonic oscillator model (dashed lines) is excellent for the fundamental, as well as for the harmonic at low intensities, when taking into account the variation in $\chi^{(2)}$ (dark red dashed line for a model without $\chi^{(2)}$ changes). The grey-shaded area indicates the saturation region. [...] (f) Maximum modulation of the nonlinear susceptibility as extracted from experiment (black crosses) against the values predicted by the anharmonic oscillator model (dashed orange line). The grey dashed line indicates the estimated maximum $\chi^{(2)}$ change of 18%.

The $\chi^{(2)}$ has been fitted from experimental data and is now shown in Fig. 2(e) next to the model. The following discussion on the fit and the divergence at high powers, at saturation, has been added on page 7:

Furthermore, as shown in Fig. 2(f), the extracted $\chi^{(2)}$ relative change with intensity (black crosses) matches very well with the anharmonic oscillator model (dashed orange line) for intensities below saturation (see Methods for details on extraction). The anharmonic model deviates from experiment at high intensities, which we attribute to higher order effects taking place in ITO as reported in literature ³³. A full modelling, requiring a higher number of fitting parameters, could potentially reveal which effects dominate at high intensities ³⁴. Nevertheless, below pump intensities of 50 GW/cm², the data demonstrates a modulation of the $\chi^{(2)}$ that can be explained through the anharmonic oscillator model.

and the fit method has been added to the Methods section on page 16:

Extraction of the nonlinear susceptibility change

With the knowledge of the linear properties of ITO and their evolution under optical pumping, a TMM code allows the computation of the surface field E_s at the air/ITO interface during modulation. Approximating the Fresnel coefficient at the harmonic frequency to be roughly

constant during modulation (see Fig. 2(b)), we can express the measured second harmonic field as $E_{SHG} \propto \chi^{(2)} E_s^2$. Then, for a modulation intensity I ,

$$\frac{E_{SHG}(I)}{E_{SHG}(I=0)} = \frac{\chi^{(2)}(I)}{\chi^{(2)}(I=0)} \times \frac{E_s^{2(2)}(I)}{E_s^{2(2)}(I=0)}$$

$$\Leftrightarrow \frac{\chi^{(2)}(I)}{\chi^{(2)}(I=0)} = \frac{E_s^{2(2)}(I=0)}{E_s^{2(2)}(I)} \times \sqrt{\frac{I_{SHG}(I)}{I_{SHG}(I=0)}}$$

where $I_{SHG}(I)/I_{SHG}(I=0)$ is the experimentally measured relative change in SHG. In Fig. 2(c,f), $\min(S/S_0)$ corresponds to $I_{SHG}(I)/I_{SHG}(I=0)$ and $\min(\chi^{(2)}/\chi^{(2)}_0)$ corresponds to $\chi^{(2)}(I)/\chi^{(2)}(I=0)$.

5) On page 6, line 170 it is stated that difference in modulation between fundamental and second harmonic is larger at low powers. Figure 2b seems to show a larger difference at higher powers.

In absolute terms, the range 20 to 50 GW/cm² is where the most gains are made by SHG modulation when compared to fundamental, however the sentence is indeed confusing and has been removed.

6) On page 4, the first order $\chi^{(1)}$ is a function of both frequency and time. I understand this to mean that formally, we average over many periods to describe the frequency dependence in time (slowly varying approximation) and therefore our time resolution is much greater than a period (~4fs). In the supplement, the rise time of the model is on the order of 1-10 fs. These two points seem to contradict each other.

The two models serve their separate purpose but are not in contradiction with each other.

Our anharmonic oscillator model follows the slowly varying approximation, and aims at describing the amplitude of the changes in material properties and is suited to replicate the results presented in Figure 2. In our experiment, as the probe has a duration of 225 fs, faster dynamics cannot be observed through modulation amplitude measurements.

The non-adiabatic model, as presented in [10.1038/s41567-023-01993-w], with a rise time on the order of 1-10 fs, is purely phenomenological and only aims at capturing these fast dynamics that cannot be observed through the slowly varying approximation, without any knowledge of the material properties.

Therefore, we use the slowly varying approximation for SHG to model the $\chi^{(2)}$ modulation contrast, whereas a non-adiabatic response of 1-10 fs is used to describe the spectral effects of the time-modulation. A combined model will require further theoretical advances beyond the scope of this work.

7) There is very little discussion about the technological implications of the study. There is a

proposed reconstruction algorithm, but it is mentioned in passing with sparse explanation. Additionally, is the $\chi^{(2)}$ modulation necessary in this application. Wouldn't the modulation of the linear field stemming from the strong modulation create the same potential for such applications?

The paragraph on the reconstruction algorithm has been extended to explain the benefits of using second order phenomena on page 9:

Thanks to the higher contrast between the modulated and unmodulated states achieved with the SHG from the suppression of the carrier frequency peak, the accessibility of efficient and affordable measurement equipment in the visible range and the overall enhanced nonlinearities, reconstruction algorithms are more powerful at the harmonic rather than fundamental level. More complex transformations can also be implemented by encoding information in both space and time, such as images with pixels along axes (space, time), and measuring the SHG as a function of delay and frequency. Performing linear regression on the SHG spectrum can then allow machine learning applications such as image classification and the creation of convolutional layers.

We also propose another application for the modulation of $\chi^{(2)}$ in the discussion on page 9:

Another potential application of the modulation of the nonlinear susceptibility in analogue computing could lie in reconfigurable image processing. For example, nonlocal metasurfaces have been used for thermal switching between edge-detection and transmission/reflection³⁶. A small layer of ITO embedded in such a metasurface will couple to the field without affecting the system's resonant properties³⁷, enabling reconfigurable edge detection through all-optical switching of SHG.

8) The inset of Figure 3d looks like there are three slits/pulses. Perhaps the left-most and right-most green boxes should be extended.

We thank the reviewer for his advice, the panel has been redesigned for clarity.

9) It would be very interesting to have a FROG trace or autocorrelation of the emitted SHG signal showing the temporal structure.

Unfortunately, due to the low amplitude of the modulated SHG signal, the probe intensity being low and SHG being further decreased by modulation, FROG or autocorrelation measurements are impossible to set in place in our setup. We are, however, working on developing further pulse analysis capabilities.

10) On page 8, it says the unmodulated SHG is suppressed. Based on figure 2b it seems modulation suppresses the SHG.

It was meant that the original carrier frequency peak is suppressed during modulation, while the frequency-shifted peak appears, which leads to a higher contrast. The sentence has been corrected for clarity on page 10:

Finally, thanks to the strong suppression of the carrier frequency peak from the original spectrum in the time-modulated signal

11) In the supplementary there is the high pump powers saturate the system within several femtoseconds resulting in an ultrafast rise time based on reference 5. First, we note that the intensities in reference 5 were 10 times that of this study. Is the 4.36 fs rise time still appropriate? Also, after that rise time the system continues to be pumped. One would then expect a plateau in the material response during the rest of the pulse (>200 fs). This is not captured by the model. Could excited state absorption also play a role here?

The amplitude of the observed oscillations relative to the unmodulated peak at the carrier frequency is directly related to the rise time of the modulation. The value of 4.36 fs has been picked for consistency with reference [17]. A longer rise time beyond 10 fs wouldn't yield the experimentally measured oscillations, which prompts us to question where the saturation level is in our system.

The saturation of the system depends on many factors: sample ENZ frequency, thickness, pump and probe beam size. In reference [5] the saturation level is reported at 100 GW/cm², but here linear measurements in Fig. 2 show that this level is lower in this new sample, at about 50-75 GW/cm². The SHG saturating faster due to its quadratic dependence on linear signal shows that saturation and fast dynamics could emerge in the conditions presented in Fig. 3(d).

12) The last sentence of page 2 states the frequency shift and broadening of SHG is not explained by the model. I am not able to find in the figures or text where this is discussed. Perhaps I missed it.

This comment on the model has been moved to the discussion of figure 3, on page 8.

However, the observed generation of new frequencies cannot be accurately described by our model and calls for further theoretical investigation of non-perturbative nonlinear optics, notably to include the effects of dispersion or saturation. A more rigorous model such as the one presented in ³⁶ could be extended in the future to account for SHG.

13) The SHG from the probe beam is perturbative due to the low intensity of the probe in the unmodulated medium. However, the field in the modulated medium is much larger due to the enhancement of field with Berreman mode. In this case can the SHG still be expanded with perturbation theory. Is $\chi^{(2)} \ll \chi^{(1)}$?

It is right that the Berreman resonance enhances the field in the unmodulated medium, and this is taken into account when reducing the probe intensity to avoid non-perturbative effects. When modulating, the Berreman mode disappears and the field is not further enhanced but only reduced (c.f. our reply to question 3). We have further updated supplementary figure 5 to reflect the change in electric field at the layer's surface during modulation on page 20:

Supplementary Figure 5. Surface field at the air/ITO interface. **(a)** Coupling spectrum of the probe electric field to the z -polarised field at the surface, in the absence of pumping (blue curve) and for pump intensities of 27 GW/cm² and 100 GW/cm². The dashed grey line indicates the original carrier frequency of the probe. **(b)** Simulated surface field amplitude as a function of delay for a pump intensity of 27 GW/cm² (pink curve) and 100 GW/cm² (dark curve) and a probe carrier frequency of 230 THz.

Some minor comments

14) Page 2 line 46, should there be a hyphen on Tin-Oxide? Should there also be one in the abstract?

There shouldn't be a hyphen, the typo has been fixed.

15) Page 2 line 53, silicon should not be capitalized.

Fixed.

16) The word "non-perturbative" on line 131 seems to be a typo. I understood this sentence to mean the probe is does not perturb the system. In nonlinear optics, non-perturbative has a different connotation.

Fixed.

17) On page 7, lines 215-216, it is unclear to me if the terms amplitude and frequency describe the pump or the probe of the SHG signal.

The sentence has been modified for clarity (now page 8):

Hence, the modulation of SHG signal in amplitude and frequency is very susceptible to changes in delay

18) Page 7 line 227 is missing a verb between time and set.

Fixed.

19) The figure S6 is labeled as S5.

Fixed.

20) The colors on the x axis tick marks for Figure S5 should be switched.

Fixed.

21) Are the alpha and beta in the definition of $f(t)$ the same as τ_r and τ_d ?

Alpha and Beta are scales respectively associated to τ_r and τ_d , with an inverse dependence.

22) In the fitting of the fast relaxation time, is the result unique?

We show a value of $\tau_d = \tau_r$ is consistent with our measurement, while a longer relaxation time wouldn't be consistent with measurements as the observed oscillations at higher frequencies would have lower amplitude. Supp. Fig. 9 shows the relaxation time is unique for a given modulation amplitude, with values of 1-10 fs range being in the experimental standard deviation. A relaxation time of 20 fs would be 2 standard deviations away. The following text has been added on page 24 to emphasize this:

Setting a longer relaxation time results in a decrease of $I_{\text{blue}}/I_{\text{red}}$ inconsistent with experimental measurements.

REVIEWERS' COMMENTS

Reviewer #2 (Remarks to the Author):

I am happy with the substantial revisions the authors introduced to their manuscript in response to the referees' comments, and I am pleased to recommend the paper for publication.

Reviewer #3 (Remarks to the Author):

I carefully checked the revised manuscript and responses to comments. The authors have answered my questions very well. Especially when I was concerned about the similarity between the experimental platform of this manuscript and their previous publication in Nature Physics, they provided clear response. Their previous work was focused more on the time modulation of linear dielectric constants, while this work is focused more on the time modulation of nonlinear coefficients. Although the experimental platform used in the study is very similar, the problems studied and the observed phenomena are indeed different, which has a certain degree of innovation. The authors also provided detailed responses to other experimental details. I have no further questions on the revised manuscript.

Reviewer #4 (Remarks to the Author):

The presented manuscript describes the modification of not only the linear susceptibility, but also the nonlinear susceptibility in a dynamically changing media. The Authors demonstrate the modulation and coherent manipulation of the second harmonic generation in a time-varying media. As the field of time-varying media in the optical regime has just recently experienced a renaissance with transparent conducting oxides taking center stage, this is both a timely and broadly interesting result. In the prior round of review, there were several serious questions posed that needed to be addressed before submission and a few minor edits. The Authors have answered those satisfactorily and I believe the manuscript is acceptable for publication.